# Pericardial Fluid Accumulates microRNAs That Regulate Heart Fibrosis after Myocardial Infarction

**DOI:** 10.3390/ijms25158329

**Published:** 2024-07-30

**Authors:** Elsa D. Silva, Daniel Pereira-Sousa, Francisco Ribeiro-Costa, Rui Cerqueira, Francisco J. Enguita, Rita N. Gomes, João Dias-Ferreira, Cassilda Pereira, Ana Castanheira, Perpétua Pinto-do-Ó, Adelino F. Leite-Moreira, Diana S. Nascimento

**Affiliations:** 1i3S—Institute for Research and Innovation in Health, University of Porto, 4200-135 Porto, Portugal; elsa.silva@i3s.up.pt (E.D.S.); f.r.c.m.costa@gmail.com (F.R.-C.); argomes@i3s.up.pt (R.N.G.); joaopedroesjml@gmail.com (J.D.-F.); cassildapereira@outlook.pt (C.P.); ana.castanheira@inl.int (A.C.); perpetua@i3s.up.pt (P.P.-d.-Ó.); 2ICBAS—Instituto de Ciências Biomédicas Abel Salazar, University of Porto, 4050-313 Porto, Portugal; 3INEB—Instituto Nacional de Engenharia Biomédica, University of Porto, 4200-135 Porto, Portugal; 4Center for Translational Medicine (CTM), International Clinical Research Centre (ICRC), St. Anne’s Hospital, 60200 Brno, Czech Republic; 5Department of Biomedical Sciences, Faculty of Medicine, Masaryk University, 62500 Brno, Czech Republic; 6Cardiovascular R&D Center, Faculty of Medicine, University of Porto, 4150-180 Porto, Portugal; up465119@g.uporto.pt (R.C.);; 7Instituto de Medicina Molecular João Lobo Antunes, Faculdade de Medicina, Universidade de Lisboa, 1649-028 Lisbon, Portugal; fenguita@medicina.ulisboa.pt; 8Center for Translational Health and Medical Biotechnology Research (TBIO)/Health Research Network (RISE-Health), ESS, Polytechnic of Porto, 4200-072 Porto, Portugal; 9Chemical and Biomolecular Sciences, School of Health (ESS), Polytechnic of Porto, 4200-465 Porto, Portugal; 10INL—International Iberian Nanotechnology Laboratory, 4715-330 Braga, Portugal

**Keywords:** myocardial infarction, pericardial fluid, fibrosis, miRNAs, miR-22-3p, cardiac fibroblasts

## Abstract

Pericardial fluid (PF) has been suggested as a reservoir of molecular targets that can be modulated for efficient repair after myocardial infarction (MI). Here, we set out to address the content of this biofluid after MI, namely in terms of microRNAs (miRs) that are important modulators of the cardiac pathological response. PF was collected during coronary artery bypass grafting (CABG) from two MI cohorts, patients with non-ST-segment elevation MI (NSTEMI) and patients with ST-segment elevation MI (STEMI), and a control group composed of patients with stable angina and without previous history of MI. The PF miR content was analyzed by small RNA sequencing, and its biological effect was assessed on human cardiac fibroblasts. PF accumulates fibrotic and inflammatory molecules in STEMI patients, namely causing the soluble suppression of tumorigenicity 2 (ST-2), which inversely correlates with the left ventricle ejection fraction. Although the PF of the three patient groups induce similar levels of fibroblast-to-myofibroblast activation in vitro, RNA sequencing revealed that PF from STEMI patients is particularly enriched not only in pro-fibrotic miRs but also anti-fibrotic miRs. Among those, miR-22-3p was herein found to inhibit TGF-β-induced human cardiac fibroblast activation in vitro. PF constitutes an attractive source for screening diagnostic/prognostic miRs and for unveiling novel therapeutic targets in cardiac fibrosis.

## 1. Introduction

Ischemic heart disease (IHD) stands as the foremost contributor to worldwide morbidity and mortality [1]. Within the spectrum of IHD, myocardial infarction (MI) emerges as the most predominant manifestation [2]. Acute MI includes ST-segment elevation myocardial infarction (STEMI) and non-ST-segment elevation myocardial infarction (NSTEMI). STEMI is characterized by acute total coronary occlusion and transmural myocardial necrosis, and it is treated using urgent reperfusion therapy. On the other hand, NSTEMI is often considered synonymous of sub-endocardial MI and typically occurs without total coronary occlusion. Studies comparing mortality between STEMI and NSTEMI patients have concluded that STEMI is associated with higher in-hospital death rates, while NSTEMI has a worse long-term prognosis [3,4]. Following MI, the cessation of blood flow results in cardiac cell death and impaired contractility [5,6]. Alongside these alterations, scar tissue is formed at the infarction site (replacement fibrosis) and in the surrounding myocardium [7,8,9]. Although this process is crucial at initial stages to avoid myocardial rupture, continuous deposition of myocardial fibrosis is deleterious by leading to a less compliant matrix, increased ventricular stiffness, impaired mechano-electric coupling of cardiomyocytes, and diminished pump function [10,11,12]. Also, structural and mechanical cues from the fibrotic ECM induce the continuous activation of resident fibroblasts as part of a feed-forward loop that contributes to persistent fibrotic deposition [13,14,15]. Controlling this sequence of events stands as an essential therapeutic objective in post-MI care and heart failure management.

One of the main challenges in achieving an in-depth characterization of the cardiac microenvironment after MI is the invasiveness and scarcity of myocardial biopsies. However, a potential source of biomolecules secreted by cardiac cells is the pericardial fluid (PF), which surrounds the heart. The content of the PF has been found to reflect a variety of cardiovascular diseases [16,17] and regulate several cardiac processes [18,19]. Owing to its close proximity to the heart and low clearance rate, the PF can also be capitalized for the precise and targeted delivery of therapeutics to the heart by exploring the intra-pericardial delivery route. As such, studies unveiling the composition of the PF and how it impacts cardiac cells in specific disease scenarios are needed to identify novel therapeutic molecules but also to establish the foreground knowledge for future clinical application.

Among other constituents, the PF has been shown to contain microRNAs (miRs), which are small RNA molecules that regulate gene expression [20] and key pathological processes such as inflammation and fibrosis in response to MI [21]. Kuosmanen et al. provided the first evidence of miR presence in the PF of patients undergoing open-heart surgery but was unable to associate specific miRs with specific pathologies [22]. Subsequent studies demonstrated specific miR content in atrial fibrillation [23,24] and arrhythmogenic right ventricular cardiomyopathy [25]. Regarding patients undergoing coronary artery bypass grafting (CABG), although the level of miR-423-5p was found to be higher in the PF when compared with serum, no differences were detected in the PF of patients with unstable angina pectoris, stable angina, or aortic stenosis [26]. Of note, several miRs known to regulate cardiac fibrosis are present in PF [22,27], namely miR-21-5p, which is known to potentiate fibrotic remodeling after injury [22,28,29,30,31,32,33]. Moreover, miR-382-3p, miR-450a-2-3p, and miR-3126-5p were identified in exosomes obtained from the PF of patients with atrial fibrillation and were reported to drive cardiac fibrosis [23,24] Overall, these studies demonstrate that miRs in the PF may be important regulators of cardiac remodeling by controlling the fibrotic deposition. Yet, the miR content of PF in response to MI is yet undisclosed.

Herein, we aimed to analyze alterations in the miR content in the PF after MI and investigate how these alterations impact the pro-fibrotic potential of this biofluid.

## 2. Results

### 2.1. Pericardial Fluid from STEMI Patients Accumulates Biomarkers of Cardiac Fibrosis

While the bioactive role of PF has been acknowledged, mechanistic understanding of how this biofluid affects cardiac cells remains limited. Here, to comprehend the function of this biological fluid in the pathological response to MI, PF and peripheral blood were collected from coronary patients undergoing CABG (Figure 1 and Appendix A). Three groups of patients were included in the study: 1—CTRL—patients with chronic stable angina and no previous history of acute coronary syndrome; 2—NSTEMI—patients with a recent (<3 months) and first non-ST-segment elevation myocardial infarction; and 3—STEMI—patients with a recent (<3 months) and first ST-segment elevation myocardial infarction. The groups were similar in terms of age, sex, cardiovascular risk factors (Table 1), and medication (Table 2) at the time of surgery. Left ventricle ejection fraction (LVEF) was decreased in NSTEMI and STEMI groups, owing to tissue damage triggered by the blockage of coronary arteries, which contrasts with the CTRL group composed of stable coronary patients.

Levels of procollagen type I carboxyl-terminal propeptide (PICP), produced by cardiac fibroblasts as a precursor of fibrillary collagen [34], were upregulated in PF when compared with plasma in all patient cohorts (Figure 2A). Moreover, the PF from STEMI patients showed higher PICP when compared with CTRL patients. Soluble suppression of tumorigenicity 2 protein (ST-2), a biomarker indicative of cardiac stress that is secreted during adverse cardiac remodeling and is correlated with tissue fibrosis [35], was increased in the PF of STEMI patients (Figure 2B). Furthermore, the levels of ST-2 in the PF of STEMI patients were inversely correlated with LVEF (Figure 2C), whereas no correlations were obtained regarding CTRL and NSTEMI patients (Appendix A).

To further understand if the PF after MI acquires a pro-fibrotic phenotype, human ventricular fibroblasts (hCFs) were stimulated in vitro with the PF from the three cohorts (Figure 2D). When exposed to the PF of STEMI patients for 24 h, hCFs showed increased levels of pro-fibrotic gene ACTA2 (Figure 2E). No differences were found in pro-fibrotic genes COL1A1 and CCN2 between groups (Figure 2E). Moreover, the PF of the three groups induced a robust activation of hCFs, as shown by the high frequency of alpha-smooth muscle actin (α-SMA)-positive cells after 4 days, whilst no differences were detected amongst groups (Figure 2F,G). Of note, the PF of NSTEMI patients induced a lower deposition of collagen type I when compared with STEMI and CTRL (Figure 2F,G). These findings support the observation that although the PF of STEMI patients concentrates fibrosis-associated molecules PICP and ST-2, no particular pro-fibrotic effect is observed in vitro when compared with the other patient groups.

### 2.2. RNA Sequencing Reveals Enrichment in miRs Associated with Extracellular Matrix (ECM) Remodeling in STEMI Patients

The miRNome of PF collected from our patient cohorts was subjected to small RNA sequencing to identify relevant candidates in the pathophysiology of MI. Of note, 275 miRs were identified in all sequenced samples (Appendix A). A network analysis performed using the miRNet 2.0 online platform highlighted association with heart failure with the lowest p-value, and clustering analysis further highlighted the most relevant miRs–cardiovascular disease interactions (Figure 3A and Appendix A). Regarding miR function, the hits relate to “Innate Immunity”, “Hematopoiesis”, and “Vascular Inflammation” (Figure 3A). Owing to the lack of prior studies using deep sequencing in PF, our results were firstly compared with the only available miR profiling, by Kuosmanen and colleagues, of PF obtained using microarrays for 742 miRs [22]. A Venn diagram summarizing these results showed that 98 miRs were identified in both studies, 7 were only identified by Kuosmanen et al., and 171 constitute newly identified miRs in the PF (Figure 3B). Of note, miR-125b-5p, miR-320b, miR340- p, miR-497-5p, miR-99b-5p, and let-7d-3p, which have been reported as specific to PF when compared with other biological fluids [22], were all present in our samples, with a coverage of 100%. When focusing on the 50 most abundant miRs present in all groups, the KEGG pathway prediction analysis identified “Extracellular matrix (ECM)-receptor interaction”, supporting the hypothesis that PF-contained miRs are possibly involved in the regulation of ECM remodeling (Appendix A). Regarding a group comparison, 15, 11, and 4 miRs were differentially regulated in NSTEMI vs. CTRL, STEM vs. CTRL, and STEMI vs. NSTEMI, respectively (Figure 3C). A KEGG pathway analysis on dysregulated miRNA-targeted genes showed “Pathways in cancer” as a common pathway altered in STEMI and NSTEMI compared with controls, whereas no significant gene ontology pathway was retrieved for the STEMI vs. NSTEMI comparison. The TGF-β signaling was one of the altered pathways in STEMI vs. CTRL (Figure 3D), further supporting that STEMI PF may regulate cardiac fibrosis. Accordingly, a functional analysis of STEMI vs. CTRL miR further highlighted “cardiac remodeling” as a main process regulated by these miRs (Figure 3E). From these, nine miRs were augmented and two were less abundant in STEMI compared with CTRL (Figure 3F). Importantly, three of the identified miRs have been associated with cardiac fibroblast activation, namely miR-125 [36], miR-21-5p [37], and miR-30e-5p [35]. Contrarily, miR-203-3p [38], miR-532 [39], miR30e-5p [35], miR-22-3p [40,41], and miR-146a-5p [42] have been linked with anti-fibrotic effects. Overall, these discoveries illustrate that PF serves as a reservoir of miRs known to modulate cardiac fibrosis bidirectionally and whose abundance is modified after acute ischemic events, such as MI.

### 2.3. miR-22-3p Overexpression Reduces hCF Activation by TGF-β

To better understand the functional relevance of the anti-fibrotic miRs in the PF of STEMI, we decided to focus on miR-22-3p. The RNA sequencing results of miR-22-3p were validated by a qPCR analysis (Figure 4A). For that analysis, and because there is limited information regarding the best miR for normalization in the PF, we resorted to RNA sequencing and bibliography and selected miR-99a-5p [39]. Moreover, we also added a defined amount of synthetic non-human miRNA, cel-miR-39-3p miRNA, to our RNA isolation. Thus, we also used the *C. elegans* miR-39-3p (miR-39-3p) to normalize our RNA sequencing results (Figure 4A). Corroborating the RNA sequencing results, miR-22-3p was increased in the STEMI group, and thus, we decided to further explore the source and role of this miR in cardiac fibrosis (Figure 4A).

The increased abundance of miR-22-3p in the PF of STEMI patients may further impact the heart. miR-22-3p is a microRNA associated with skeletal and cardiac muscle function, and it has been identified in cardiomyocytes [43,44]. However, the extent to which other cell types contribute to the expression of this miR, particularly in humans, remains unclear. To address this, we investigated miR-22-3p expression in the main cell types composing the human heart, i.e., embryonic pluripotent-derived ventricular cardiomyocytes (HESC-CMs), cardiac microvascular endothelial cells (hECs) and hCFs (Appendix A). Of note, all analyzed cells expressed miR-22-3p, indicating that the main cardiac cell lineages constituting the human heart contribute to miR-22-3p production.

To evaluate whether the accumulation of miR-22-3p around the heart exerts an influence on the cardiac tissue, we explored the impact of miR-22-3p overexpression on hCFs in the presence and absence of TGF-β (Figure 4B). Transfection was optimized by comparing the efficiency of 100 nM and 200 nM of mir-22-3p or control (scrambled) precursors (Figure 4C). Overexpression was evidently attained with both conditions; therefore, the lowest concentration was selected for the following experiments. Consistent overexpression of miR-32-3p was obtained on day 2 (8,3-fold) and 6 (18-fold) post-transfection (Figure 4D). To understand the role of miR-22-3p on hCF activation, transfected cells were then stimulated with TGF-β (Figure 4B). Upon treatment, miR-22-3p-overexpressing hCFs displayed a downregulation of pro-fibrotic genes ACTA2 and COL1A1 when compared with the precursor control (Figure 4E). Importantly, a downregulation of markers of myofibroblast differentiation α-SMA and COL I was verified upon overexpression of miR-22-3p (Figure 4F,G). Cells exhibited a fusiform elongated morphology, retaining a non-activated fibroblast morphology after miR-22-3p overexpression. These results collectively suggest a protective role of miR-22-3p in the TGF-β-mediated hCF activation, positioning it as a potential therapeutic candidate for managing cardiac fibrosis in the context of MI. Altogether, these findings demonstrate that the PF is altered upon MI, and it accumulates miRs and other fibrosis-related molecules that contribute to the formation of a pro-fibrotic environment. Notwithstanding, the PF of STEMI patients also accumulates anti-fibrotic miRs, such as miR-22-3p, counter-balancing the pro-fibrotic effect of other miRNAs in the PF by inhibiting cardiac fibroblast activation in response to TGF-β (Figure 5).

## 3. Discussion

The aim of this study was to evaluate the impact of PF on cardiac fibroblast activation and identify altered miR content in the PF of STEMI patients for potential integration in anti-fibrotic therapies for MI.

We demonstrated that PF accumulates higher concentrations of the fibrotic molecules PICP and ST-2, with the latter being correlated with systolic dysfunction in STEMI. The PF of all cohorts showed a similar ability to induce cardiac fibroblast activation in vitro. The PF from STEMI patients displays an accumulation of miRs associated with pro- and anti-fibrotic effects. From those miRs, miR-22-3p was found to be downregulated in the myocardium after MI. Moreover, this miR inhibited TGF-β-induced human cardiac fibroblast activation in vitro.

PF content has previously been shown to become altered with cardiac diseases [17,45], constituting a potential source of disease biomarkers and/or therapeutic targets. One could hypothesize that the concentration and molecular composition of the PF may reflect ongoing ventricular remodeling following an acute event. The correlation between ST-2 levels in PF and reduced LVEF in STEMI patients is aligned with this premise and previous literature findings [46,47]. In coronary heart disease patients, the levels of growth differentiation factor 15 (GDF-15) in PF are correlated with LVEF, biomarkers of renal dysfunction, and inflammatory molecules [48]. GDF-15 was also elevated in patients with MI, in which it was associated with an increased risk of mortality during the 1-year follow-up period [49]. In our work, the levels of PICP and ST-2 in PF were higher when compared with plasma levels, which may indicate a more accurate representation of the heart’s pathophysiology [50,51]. Altogether, these studies further support the relevance of using the PF as a valuable source of biomarkers for identifying disease severity and predictors of cardiovascular complications.

Given that PF serves as a reservoir for molecules produced by heart cells, it is logical to speculate that, under pathological and physiological conditions, PF may impact cardiac cells in a feedback loop. In our work, we demonstrated that the PF from the three cohorts induced cardiac myofibroblast differentiation, although there was more pronounced Collagen type I production in STEMI patients. Previously, the PF was shown to affect endothelial cells; namely, the PF from patients undergoing CABG promote higher vasoconstriction of mesenteric arteries through an endothelin-1-based mechanism [52]. Endothelin-1 is a potent vasoconstrictor that has been involved in processes that leads to endothelial dysfunction [53]. In the opposite direction, extracellular vesicles isolated from PF of aortic stenosis patients were shown to promote therapeutic angiogenesis in a mouse model of ischemia as well as improve the survival, networking, and proliferation of cultured endothelial cells [18]. These findings and the herein work support PF as an engaged regulator of cardiac activity rather than as a passive participant by serving as a repository for both protective and harmful molecules. Furthermore, the impact of PF on other cardiac cells such as cardiomyocytes and even in intercellular communication involving, for instance, cardiomyocytes and cardiac fibroblasts is further needed and can be achieved by resorting to human-induced pluripotent stem cells [54]. Investigating this combination may unveil crucial molecular signals pertinent to the pathophysiology of MI.

The analysis of the PF miRNome by RNA sequencing further corroborated that PF reflects the content of the heart through the 275 identified miRs present in all samples, where the highest scores are associated with heart failure [22]. This analysis further strengthened that PF may influence the cardiac extracellular compartment, considering that the 50 most abundant miRs in the PF were mostly associated with ECM interactions in the GO analysis. Kuosmanen et al. found no differences in miR content among various heart diseases but identified several abundant miRs in PF, including miR-21-5p, miR-451a, miR-125b-5p, let-7b-5p, and miR-16-5p [22], which were among the most abundant miRs in our analysis. Focusing on STEMI versus CTRL patients, we identified 11 miRs linked to TGF-β pathway regulation. Among these, eight miRs have been previously associated with cardiac fibrosis [35,42,55,56]. Within those miRs, miR-21 is well described in promoting cardiac fibrosis in murine models of pressure overload and MI [37,57]. In these models, inhibition of miR-21 mitigated cardiac fibrosis and dysfunction [57]. In human failing hearts, inhibition of mir-21 has cardioprotective effects, namely by decreasing the expression of genes associated with cardiac fibrosis and fibroblast activation [58].

In the opposite direction, miR-203-3p has been shown to mitigate myocardial fibrosis and oxidative stress in murine models of diabetic cardiomyopathy through the PI3K/Akt signaling pathway [37]. Moreover, overexpression of miR-203-3p in mouse cardiomyocytes led to a downregulation of fibrotic genes such as TGF-β1, connective tissue growth factor (CTGF), and fibronectin (FN) [38]. The accumulation of miRs regulating cardiac fibrosis in the PF of STEMI patients further supports the use of this biofluid as a source of biomarkers and as a model to indirectly assess myocardial extracellular remodeling.

We further focused our attention on miR-22-3p once the role of this miR in the activation of human cardiac fibroblasts had not been previously investigated, despite it being advanced as a predictor of coronary artery disease [59] and being demonstrated to be up-regulated in the plasma of MI patients [60,61]. In line with this evidence, we demonstrated elevated levels of miR-22-3p in the PF of the STEMI patients, whose ischemic damage is considerably superior to CTRL and NSTEMI.

We further demonstrated that miR-22-3p overexpression in hCFs inhibited fibroblast-to-myofibroblast differentiation. Indeed, in murine cardiac fibroblasts, miR-22-3p was found to protect CF from TGF-β-mediated activation [40], a mechanism that is at least in part explained by the direct inhibition of TGF-βR1. This aligns with previous research showing that miR-22-3p overexpression reduces cell migration and collagen deposition and inhibits the proliferation of mouse cardiac fibroblasts upon angiotensin-II treatment [52] through direct inhibition of the platelet-activating factor receptor (PTFAR) [62]. Moreover, inhibition of miR-22-3p potentiates fibrogenesis in cultured murine cardiac fibroblasts [40]. However, conflicting findings further suggest that miR-22-3p overexpression may promote senescence and fibroblast activation, contributing to cardiac fibrosis in aging hearts [63]. It is important to note that in the work conducted by Hong et al., adult mice were used, which differs from the aged mouse model employed in the latter study [40]. Thus, it is evident that aging induces alterations in the cardiac microenvironment, leading to changes in the behavior of heart cells, including cardiac fibroblasts. Moreover, these changes extend to the responsiveness of these cells to various stimuli, namely in chronic activated fibroblasts in which the TGF-β signaling plasticity is reduced and cannot be reversed to a quiescent state [64]. Timing may be crucial, as miR-22-3p’s role seems to vary with environment and stimuli.

The main strength of this study is unveiling alterations in the PF of human subject in response to MI for the first time and showing the in vitro anti-fibrotic potential of miR-22-3p, an miR enriched in the PF of STEMI patients. However, a significant limitation of this study is the small sample size, which was all collected from a single hospital. As such, future multicenter studies using larger cohorts are necessary to increase the reliability and generalizability of our findings. Another limitation of this study is the present lack of knowledge on the miR composition of PF from healthy individuals to be used as a reference control. In fact, the collection of PF from healthy patients is not possible because PF collection requires access to the pericardial space, only possible during open-chest surgery. As such, herein, stable coronary patients without a previous history of MI were used as a control, which allows for the identification of miR alterations upon acute MI but limits conclusions regarding the interest of the identified miRNA as biomarkers of coronary artery disease.

Our study combining RNA sequencing with in vitro models reveals that the PF of STEMI patients has molecules with dual effects on fibrogenesis and contains miRs regulating cardiac fibrosis, including miR-22-3p, which seems to be protective against hCF activation. In fact, this miRNA may be impacting other cardiac cells as it has been shown to promote cardiomyocyte hypertrophy and regulate sarcomere organization and metabolic shift during cardiac remodeling [65,66].

Altogether, our study demonstrates that the content of the PF changes after MI and that STEMI patients accumulate a plethora of miRs that are able to regulate cardiac fibrosis. Specifically, miR-22-3p may impact the heart in a feedback loop by buffering the pro-fibrogenic potential of other miRs enriched in this biofluid after an extensive MI. As such, this study contributes to increasing our knowledge on the composition and modulatory role of PF after MI. This is particularly relevant to guide future interventions as the development of novel delivery technologies for non-invasive access to the pericardial sac in recent years [67] has allowed for the exploitation of intra-pericardial delivery as an alternative route for local delivery of novel therapeutic agents and bioengineered solutions to the heart.

## 4. Materials and Methods

### 4.1. Study Participant Details

This study was performed using the following patient cohorts undergoing coronary artery bypass grafting (CABG): (i) a control, featuring stable coronary artery patients (stable angina, without previous MI or acute coronary syndrome); and (ii) the two MI cohorts, the NSTEMI cohort (composed of patients with a recent (<3 months) and first non-ST-segment elevation myocardial infarction) and the STEMI cohort (composed of patients with a recent (<3 months) and first ST-segment elevation myocardial infarction). The groups were characterized in terms of age, sex, cardiovascular risk factors, and medication at the time of surgery (Table 1 and Table 2).

### 4.2. Sample Collection and Processing

Peripheral blood (9–10 mL, EDTA being used as an anti-coagulant) and pericardial fluid (PF) (2–10 mL, depending on the patient PF volume) were collected and processed within two hours after collection (Appendix A). Plasma and PF were centrifuged at 1200× *g* for 10 min (min) at room temperature (RT) without braking and acceleration, the liquid fraction was collected and centrifuged twice at 2500× *g* for 15 min at 4 °C, and the liquid fraction was stored at −80 °C.

### 4.3. Cardiac Fibrosis Markers Quantification

An enzyme-linked immunosorbent assay (ELISA) kit was used to detect and quantify human procollagen type I carboxy-terminal propeptide (PCIP) and human interleukin (IL)-1 receptor 4/suppression of tumorigenicity 2 (ST-2) (RayBio^®^, Peachtree Corners, GA, USA) levels in the PF and plasma of patient cohorts according to the manufacturer’s instructions. The absorbance was read at 450 nm.

### 4.4. Cell Culture

Primary adult human cardiac fibroblasts (hCFs; Cell Applications, Inc., San Diego, CA, USA), embryonic derived stem cells (StemCell Technologies, Vancouver, BC, Canada), and cardiac microvascular endothelial cells (hECs) (Lonza, Hyderabad, India) were cultured (passage 2 to 7) in fibroblast growth medium (FGM; Cell Applications, Inc.), an EGM blue kit (Lonza), and an MTER kit (StemCell Technologies), respectively. Cells were incubated in a HERAcell^®^ 150 CO_2_ incubator (Heraeus^®^, Hanau, Germany) at 37 °C in a humidified atmosphere containing 5% CO_2_.

### 4.5. Human Embryonic Stem Cells Differentiation into Ventricular Cardiomyocytes

Embryonic stem cells (StemCell Technology) were differentiated into ventricular cardiomyocytes using a STEMdiff™ ventricular cardiomyocyte differentiation kit according to the manufacturer’s instructions.

### 4.6. Human Cardiac Fibroblasts Culture with PF from Patients

hCFs were seeded at a density of 36,500 cells/cm^2^ and incubated in seeding medium (Dulbecco’s modified Eagle’s medium high glucose (DMEM; Thermofisher, Waltham, MA, USA) supplemented with 10% fetal bovine serum (FBS; Lonza), 1% penicillin-streptomycin (P/S; Biowest, Nuaillé, France), and 0.2 mM 2-phospho-L-ascorbic acid (Asc-2P; Sigma Aldrich, Co., St. Louis, MO, USA) for 24 h. Then, hCF media were changed to working medium (WM) (high-glucose DMEM supplemented with 0,1% FBS, 1% P/S, and 0.2 mM Asc-2P. After 48 h in WM, the cell medium was replaced by WM containing 2% PF. Please check that intended meaning has been retained.The medium was changed every 48 h.

### 4.7. Transfection of Human Cardiac Fibroblasts

hCFs were seeded at density of 15,000 cells/cm^2^ in FGM during 48 h. Then, hCFs were incubated during 6 h with a mix of OptiMEM containing lipofectamine (RNAiMax, Thermofisher™) and the precursor miRNA hsa-miR-22-3p (Life Technologies^TM^, Carlsbad, CA, USA) according to the manufacturer’s instructions. The pre-miR™ miRNA precursor negative control (Thermofisher™) was used as the control. After incubation, cells were left overnight in FGM. Subsequently, the medium was replaced by WM either with or without 10 ng/mL TGF-β. The medium was changed every 48 h.

### 4.8. RNA Extraction and Reverse Transcription

#### 4.8.1. miR Extraction from PF and Plasma and Reverse Transcription

RNA extraction from plasma and PF samples was performed using the miRNeasy^®^ Serum/Plasma Kit (QIAGEN, Hilden, Germany), following the manufacturer’s instructions. RNA quantity and quality were assessed using a NanoDropTM 1000 spectrophotometer (ThermoFisher Scientific). RNA was stored at −80 °C. For complementary DNA (cDNA) production, the miScript^®^ II RT kit (QIAGEN) was used following the manufacturer’s instructions, and cDNA was stored at −20 °C.

#### 4.8.2. RNA Extraction and Reverse Transcription from Cultured Cells

mRNA extraction and reverse transcription for synthetizing cDNA from cultured hCFs were performed using the QIAzol reagent (QIAGEN) and PrimeScript RT reagent kit (Takara, Beijing, China), respectively, following manufacturer instructions.

MiR extraction from cultured cells was performed using the miRNeasy^®^ Mini Kit (QIAGEN), RNeasy^®^ Plus Mini (QIAGEN), or RNeasy Mini Kit (QIAGEN) according to the manufacturer’s instructions. Reverse transcription was performed using the miRCURY^®^ RT Kit (QIAGEN) according to the manufacturer’s instructions. RNA was stored at −80 °C and cDNA was stored at −20 °C.

### 4.9. Real-Time PCR (RT-PCR)

To detect miRs, a qRT-PCR analysis was performed using the miScript^®^ SYBR Green PCR Kit (QIAGEN) and miRCURY^®^ SYBR Green PCR Kit (QIAGEN). To quantify mRNA expression, the qRT-PCR analysis was performed using an iTaqTM Universal SYBR Green Supermix (Bio-Rad, Hercules, CA, USA) according to the manufacturer’s instructions on a CFX96™ Real-Time PCR Detection System (Bio-Rad, Hercules, CA, USA). Primer sequences are available in Appendix A.

### 4.10. RNA Sequencing (RNA-Seq) Analysis

miRs were extracted from the PF of control (CTRL) and MI cohorts as described above. The quality of RNA was assessed using a 2100 BioAnalyzer (Agilent, Santa Clara, CA, USA). The preparation of miR-seq libraries from 4 ng of RNA samples and subsequent RNA sequencing (RNA-seq) were performed by the GeneCore Sequencing Facility (EMBL) using Illumina Hiseq 2500. The quality of raw data were checked by Fastqc software, version 0.11.9 and filtered by removing adaptor sequences. Read normalization was performed using the Chimira web-based system version 1.0 [68]. miRNAs were detected by aligning the filtered reads with miRbase 22.1 human mature sequences using sRNAtoolbox software (https://arn.ugr.es/srnatoolbox/, accessed on 5 June 2024). Differentially expressed miRNAs were detected by the DESeq2 algorithm.

Then, to gain functional insights, an miRNA regulatory network analysis was performed using miRNet 2.0 [66]. Heatmaps were generated using the Heatmapper web server [66,69].

### 4.11. Immunofluorescence Assay

hCFs were fixed with 4% paraformaldehyde, and then cells were permeabilized with 0.1% Triton X-100 for 5 min. Samples were blocked with 1% BSA + 4% FBS and incubated with the primary antibodies α-smooth muscle actin (SMA) (Sigma) and collagen type I (Rockland Immunochemicals, Inc., Pottstown, PA, USA) and respective secondary antibodies. Cell nuclei were stained with 4′,6′-diamino-2-fenil-indol (DAPI; 5 µg/mL). Images were acquired using a high-content fluorescence microscope (IN Cell 2200, GE Healthcare, Freiburg, Germany), and the analysis was performed using ImageJ software 1.54h or Cell Profiler software 3.1.5 (Broad Institute, Cambridge, MA, USA).

### 4.12. Statistical Analysis

Statistical testing was performed using GraphPad^®^ Prism 8.0 software. Kolmogorov–Smirnov normality or Shapiro–Wilk tests were used to evaluate the normal distribution of the data. Normally distributed data were tested with an independent sample Student’s *t*-test and one-way ANOVA (Bonferroni post-hoc test) test for two or three groups, respectively. Outliers were excluded by the ROUT analysis. Non-normally distributed data were tested with the Mann–Whitney U test and Kruskal–Wallis one-way ANOVA test for two or three groups, respectively. The results are presented as a box plot with min/max whiskers or as column bars with the means ± SEM. The differences between groups are considered significant when *p* < 0.05.

The correlations between left ventricle ejection fraction (LVEF) and levels of ST-2 and PCIP in plasma and PF were assessed using a Spearman test.

## Figures and Tables

**Figure 1 ijms-25-08329-f001:**
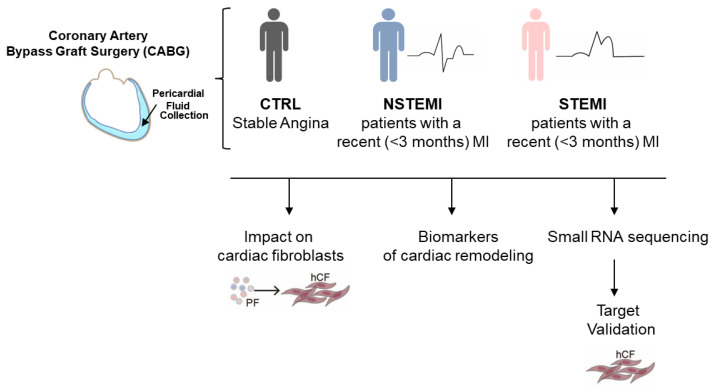
Schematic representation of the study design. Pericardial fluid (PF) was collected from coronary patients undergoing CABG and subjected to an in vitro function assay to evaluate cardiac fibroblast activation and conduct a quantification of fibrosis-associated biomarkers and small RNA sequencing. The selected targets were validated and tested in vitro with respect to hCF activation.

**Figure 2 ijms-25-08329-f002:**
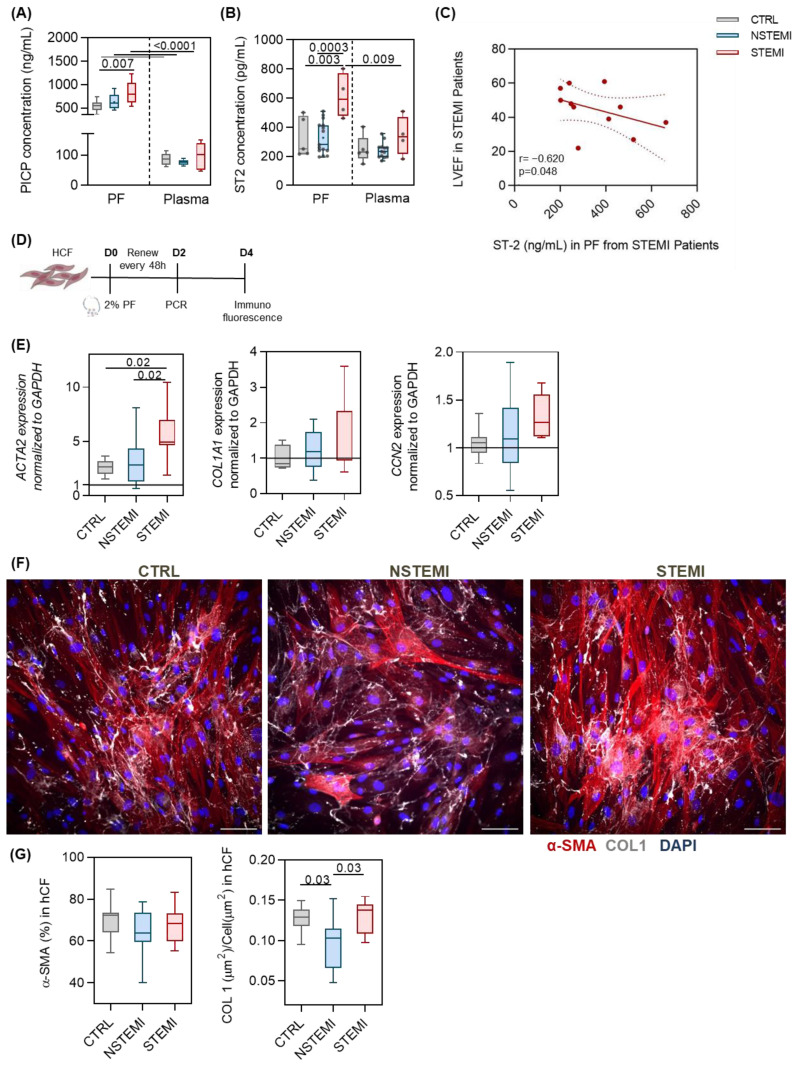
Pericardial fluid is a reservoir of fibrosis-related molecules and induces cardiac fibroblast activation. (**A**,**B**) Quantification of procollagen I C-terminal propeptide (PICP) and serum stimulation-2 (ST-2) in the pericardial fluid and plasma from the patient cohort (*n* ≥ 5/group). (**C**) Spearman correlation between left ventricle ejection fraction (LVEF) and ST-2 levels in the PF of STEMI patients (*n* = 11/group). (**D**) Schematic representation of the experimental design. (**E**) Expression of ACTA2, COL1A1, and CCN2 in hCFs cultured in 2% pericardial fluid (*n* ≥ 7/group). (**F**,**G**) Representative images and respective quantification of the alpha-smooth muscle actin (α-SMA) and collagen type I (COL I) of hCFs cultured in 2% pericardial fluid.

**Figure 3 ijms-25-08329-f003:**
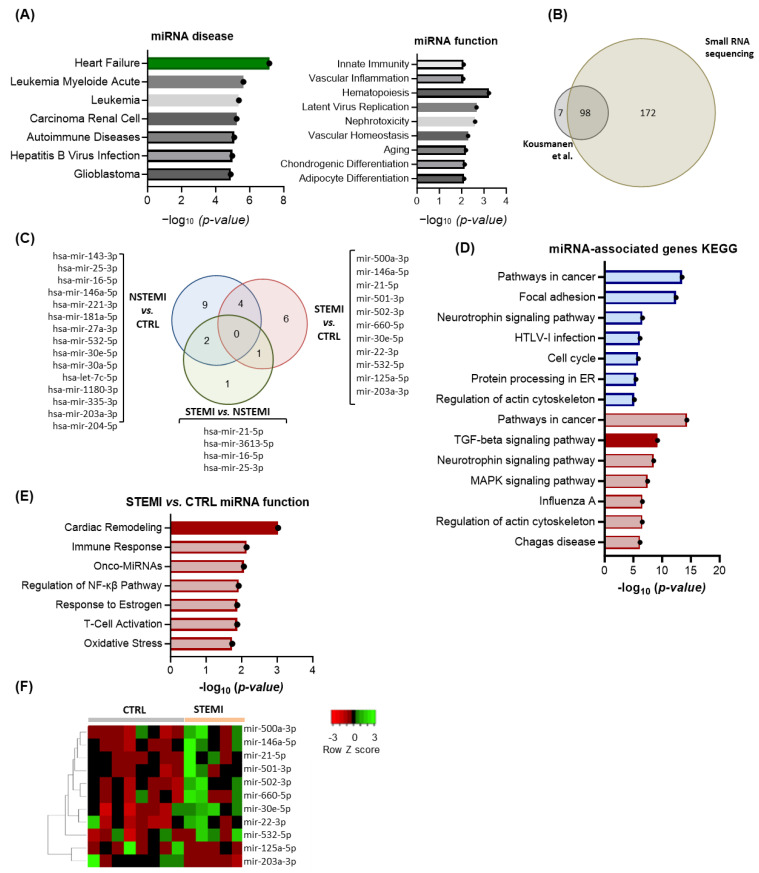
Small RNA sequencing of PF collected from CTRL, NSTEMI, and STEMI patients undergoing CABG. miRNAs identified in all PF samples by small RNA sequencing (*n* = 8, CTRL; *n* = 9, NSTEMI; *n* = 5, STEMI). (**A**) Enrichment analysis (miRNet 2.0) of miRNA–function and miRNA–disease (based on TAM 2.0) of the 275 miRs present in all groups. (**B**) Venn diagram with the comparison between our miRs list, with miRs reported by Kuosmanen et al. [22]. (**C**) Venn diagram with the number and intersection of miRs with statistically different abundance in NSTEMI vs. CTRL, STEMI vs. CTRL, and STEMI vs. NSTEMI. (**D**) Enrichment analysis of the KEGG pathways of miR gene targets regulated by altered miRNAs in STEMI patients when compared with CTRL. In red, the KEGG pathways relevant for cardiac fibrosis. (**E**) Enrichment analysis (miRNet 2.0) of miR function associated with altered miRs in STEMI patients when compared with CTRL (based on TAM 2.0). (**F**) Heatmap of altered miRNAs in STEMI patients when compared with CTRL.

**Figure 4 ijms-25-08329-f004:**
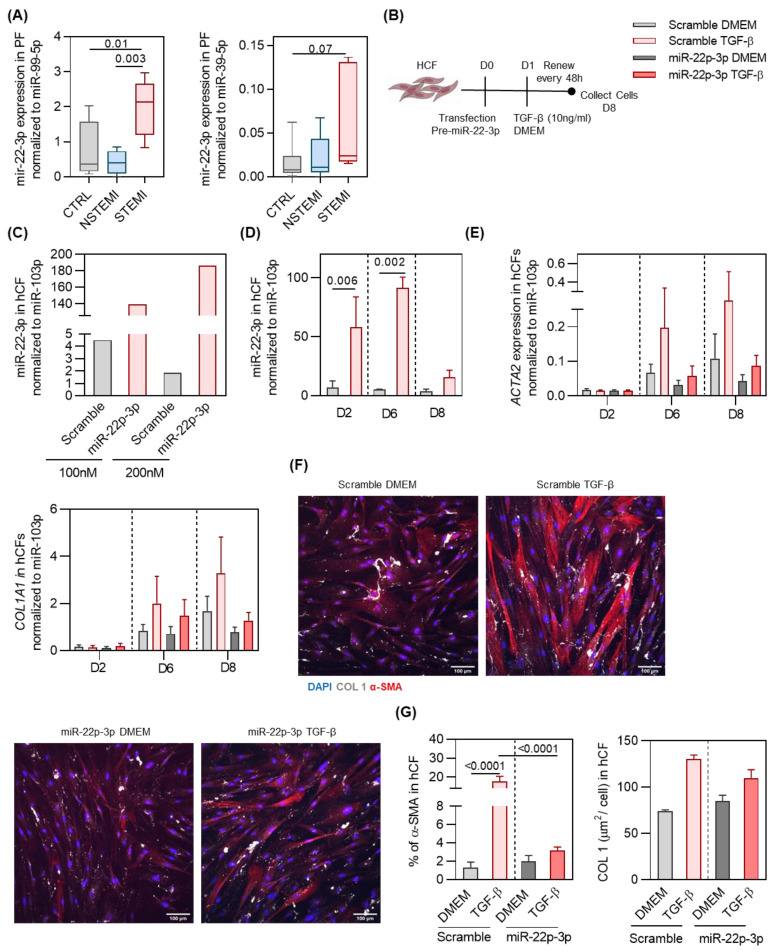
miR-22-3p overexpression inhibits hCF activation in vitro. (**A**) qRT-PCR validation of the RNAseq results for mir-22-3p and miR-203-3p. Expression is presented through the 2-ΔCt method, with normalization of the expression of miR-99-p and miR-39-5p (*n* ≥ 5/group). (**B**) Schematic representation of the experimental design. (**C**) Expression of miR-22-3p in hCFs after transfection with 100 or 200nM of miR-22-3p or scramble miR precursor. (**D**) Expression of miR-22-3p in hCFs at 2, 6, and 8 days after transfection with miR-22-3p or scramble miR precursor. (**E**) Expression of ACTA2 and COL1A1 in hCFs at different time points after transfection (*n* = 2/group). (**F**) Representative figures of α-SMA and COL I in transfected hCFs following a 7-day activation assay. (**G**) Percentage of α-SMA + cells (*n* = 4/group) and area of COL I per cell (*n* = 2/group).

**Figure 5 ijms-25-08329-f005:**
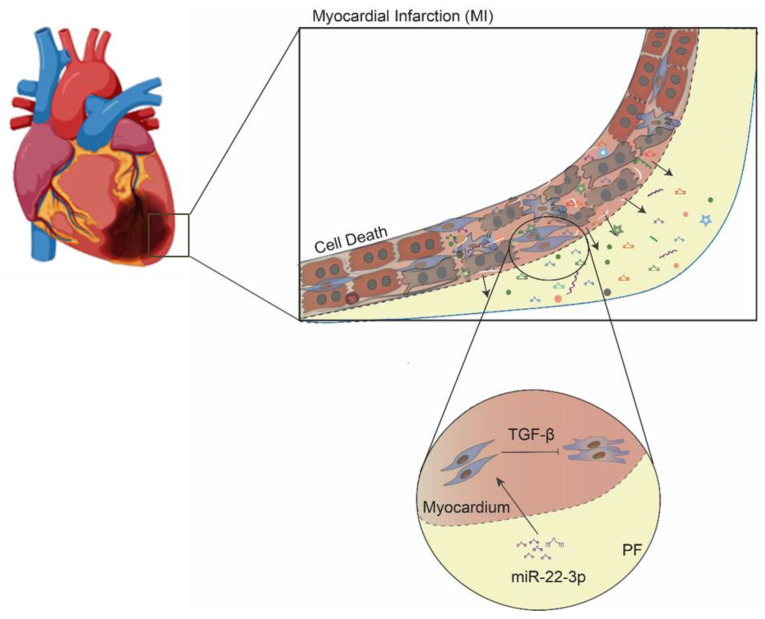
A model for the role of miR-22-3p in cardiac fibrosis after MI. Several miRs previously shown to regulate pro-fibrotic or anti-fibrotic mechanisms were released, and they accumulate in the PF after MI. From those, miR-22-3p was shown to mitigate TGF-β-induced cardiac fibroblast activation in vitro, which may counterbalance the pro-fibrotic environment formed after MI in vivo.

**Table 1 ijms-25-08329-t001:** Patients cohorts are divided into 3 groups: the control group (CTRL), comprising patients with stable coronary disease and no record of previous acute coronary syndrome; and the two myocardial infarction (MI) cohorts: the NSTEMI cohort (composed of patients with a recent (<3 months) and first non-ST-segment elevation myocardial infarction) and the STEMI cohort (composed of patients with a recent (<3 months) and first ST-segment elevation myocardial infarction). NS: Non-significant.

	CTRL (*n* = 9)	NSTEMI (*n* = 25)	STEMI (*n* = 8)	ANOVA *p*-Value	Dunn’s Multiple Comparisons Test *p*-Value		
					CTRL vs. NSTEMI	CTRL vs. STEMI	NSTEMI vs. STEMI
Age (years)	65.6 ± 8.8	68.8 ± 7.1	62.2 ± 9.2	NS	NS	NS	NS
Female (%)	1 (11)	1 (4)	1 (13)	NS	NS	NS	NS
LV EF (%)	55.75 ± 5.97	45.04 ± 13.07	44.75 ± 10.74	0.005	0.06	0.14	NS
IHD Risk Factors							
BMI (Kg·m^−2^)	28.23 ± 2.56	27.50 ± 4.10	26.89 ± 4.00	NS	NS	NS	NS
Diabetes mellitus (%)	4 (44)	13 (52)	5 (62)	NS	NS	NS	NS
Hypertension (%)	5 (56)	21 (84)	6 (75)	NS	NS	NS	NS
Dyslipedemia (%)	8 (89)	18 (72)	7 (88)	NS	NS	NS	NS
Smokers (%)	1 (11)	4 (16)	2 (25)	NS	NS	NS	NS
Ex-smokers (%)	5 (56)	10 (40)	4 (50)	NS	NS	NS	NS
Previous IHD (%)	0 (0)	4 (16)	1 (13)	NS	NS	NS	NS

**Table 2 ijms-25-08329-t002:** Patient cohorts’ medication data at the time of the surgery. NS: Non-significant.

	Control (*n* = 9)	NSTEMI (*n* = 25)	STEMI (*n* = 8)	ANOVA *p*-Value	Dunn’s Multiple Comparisons Test *p*-Value		
					CTRL vs. MI	CTRL vs. MI	CTRL vs. MI
Aspirin (%)	6 (66.66)	16 (64.00)	5 (62.50)	NS	NS	NS	NS
β-blockers (%)	6 (66.66)	14 (56.00)	3 (37.50	NS	NS	NS	NS
Statins (%)	7 (77.78)	16 (64.00)	8 (100.0)	NS	NS	NS	NS
ACEI (%)	3 (33.33)	9 (36.00)	2 (25.00)	NS	NS	NS	NS
Antidiabetics (%)	3 (33.33)	11 (44.00)	6 (75.00)	NS	NS	NS	NS
Fibrates (%)	0 (0.00)	0 (0.00)	1 (12.50)	NS	NS	NS	NS
Ca^2+^ Channel Blockers (%)	5 (55.55)	5 (20.00)	0 (0.00)	NS	NS	NS	NS
Antiaggregant (%)	1 (11.11)	6 (24.00)	2 (25.00)	NS	NS	NS	NS
Nitrates (%)	3 (33.33)	9 (36.00)	1 (12.50)	NS	NS	NS	NS
ARB (%)	1 (11.11)	7 (28.00)	0 (0.00)	NS	NS	NS	NS
Spironolactone (%)	0 (0.00)	3 (12.00)	0 (0.00)	NS	NS	NS	NS
Bronchodilators (%)	2 (22.22)	3 (12.00)	0 (0.00)	NS	NS	NS	NS
Anticoagulants (%)	1 (11.11)	3 (12.00)	1 (12.50)	NS	NS	NS	NS
Corticosteroids (%)	1 (11.11)	2 (8.00)	0 (0.00)	NS	NS	NS	NS

## Data Availability

The original contributions presented in this study are included in the article/Appendix A; further inquiries can be directed to the corresponding author. This paper does not report the original code. Any additional information required to reanalyze the data reported in this paper is available from the lead contact upon request.

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
