# Peer review of "Pericardial Fluid Accumulates microRNAs That Regulate Heart Fibrosis after Myocardial Infarction"

_ijms, 2024, doi:10.3390/ijms25158329_

Round 1

Reviewer 1 Report (Previous Reviewer 3)

Comments and Suggestions for Authors

I thank the authors for the improvements and comments that they made on this manuscript. While many points that were unclear could be corrected it remains still unclear how this observation can be translated to improved patients handling as suggested by the authors. It is puzzling that the main finding of this study is that STEMI patient’ PF gave in vitro more collagen-1 by less TGF activation. This is difficult to understand at best. For diagnostic purposes it remains unclear why miR analysis is superior to ST-2 quantification. For treatment it is unclear how you plan to modify the action of miRs in PF. Overall the observation leads to many speculations but no clear strategy that could be proven by an experiment.  

Author Response

The aim of this study was to perform an exploratory study to identify in the PF miRs that could be of theragnostic interest in the context of MI, with a particular relevance on the context of cardiac fibrosis. The authors report that the PF from CTRL, NSTEMI and STEMI show similar levels of cardiac fibroblast activation. RNA-seq analysis further identified miRs dysregulated in the PF from STEMI that were associated with a pro- and anti-fibrotic effect. From those, miR-22-3p was selected and its role on the TGF-b-induced fibroblast activation was shown in vitro, advancing this miR as a putative target for future studies validating it theragnostic potential. In addition, this study deepens our comprehension of PF composition and its impact on post-MI modulation.

We truly believe that our description of collagen production after stimulation of CF with the PF oof the three cohorts was not accurate and not contributing to a clear message. In fact, there is similar activation levels of CF activated by the PF of NSTEMI, STMI and CTRL, when considering SMA positive cells. Regarding collagen production, PF from NSTEMI induces a decrease in collagen type I, whereas no difference is observed between CTRL and STEMI. We have no clarified and corrected our description of this data which we believe was inducing confusion and was at the basis of the Reviewer comment.

Intrapericardial administration route is under exploitation for therapeutic purposes, either for delivery of drugs, cells, vectors and biomaterials for cardiac repair, this holds significant relevance for future interventions, especially considering recent advancements in delivery technologies enabling non-invasive access to the pericardial sac. As such, regarding the “plan to modify the action of miRs in PF“, one could envisage either the intrapericardial delivery of vectors for myocardial modulation of miR production or directly delivery of anti-miRs for miR silencing in vivo. A potential challenge lies in ensuring effective delivery while avoiding removal by the pericardial lymphatic system, however, a similar strategy has been already proven successful in porcine models (DOI: 10.1038/gt.2011.52).

Reviewer 2 Report (New Reviewer)

Comments and Suggestions for Authors

To:

Editorial Board

Title: “Pericardial fluid accumulates microRNAs that regulate heart fibrosis after myocardial infarction”

Dear Editor,

I read this manuscript and I think that:

-       The abstract should be revised. The results section of the abstract is poorly written as it reported no numerical data. It is difficult to follow the content of this fundamental part of the manuscript. Please rewrite.

-       Introduction section anticipated the methods section by including information about the design of the study which is beyond the aims of the introduction section. Authors should remove the sentences related to the divisions in groups and better focus on the aims of the study.

-       Similarly, introduction section allowed the display of anticipations related to the results of the study. This is unusual and not in line with the aims of the introduction section. Please remove the related sentences from this part of the manuscript.

-       Unfortunately, there is no control group for this study. How might authors compare their results with healthy population? How can miRNAs impact on overall prognosis in “CAD” patients as compared to “healthy” individuals? The answer to these questions is difficult to be understood.

-       The small sample size is a great limitation of this paper. Please discuss such a point in a dedicated limitation section.

-       Furthermore, a post-hoc sample size calculation should be provided.

-       Data related to CABG procedure should be discussed. ST2, for instance, is also an inflammatory marker as it is related to IL-33 (see also Ciccone MM et al. Molecules. 2013 Dec 11;18(12):15314-28): how can the time of extracorporeal circulation impact on the aims of the study? And the type of surgery? Please discuss all of these aspects.

-       Decimals deserve dots rather than virgules. Please revise the entire manuscript.

Comments on the Quality of English Language

the English should be revised by a native English speaker.

Author Response

-The abstract should be revised. The results section of the abstract is poorly written as it reported no numerical data. It is difficult to follow the content of this fundamental part of the manuscript. Please rewrite.

In accordance with the Reviewer suggestion, the Abstract of the revised manuscript has been altered to improve clarity and correct grammatical errors, particularly in the results section of the abstract.

-Introduction section anticipated the methods section by including information about the design of the study which is beyond the aims of the introduction section. Authors should remove the sentences related to the divisions in groups and better focus on the aims of the study.

In accordance with the reviewer's suggestions, the introduction was altered in order to focus more in the aim of the study and sentences related to the study design and results were removed.

-Similarly, introduction section allowed the display of anticipations related to the results of the study. This is unusual and not in line with the aims of the introduction section. Please remove the related sentences from this part of the manuscript.

We sincerely appreciate the valuable feedback from the Reviewer, the introduction was reformulated to remove the results of the study.

-Unfortunately, there is no control group for this study. How might authors compare their results with healthy population? How can miRNAs impact on overall prognosis in “CAD” patients as compared to “healthy” individuals? The answer to these questions is difficult to be understood.

We agree that these are interesting questions however, the authors believe that they fall out of the scope of the present study which was primarily designed to address the differences in miRNA content specifically after MI. Unfortunately, the collection of PF requires an invasive procedure that is not possible to execute in healthy individuals. In our study, the control group was part of a surgical cohort composed of patients with stable angina and no previous history of MI. The comparison of this group with STEMI and NSTEMI patients allows the investigation of miRs that are altered in the PF after MI in these surgical cohorts. We truly believe that, and considering the limitations associated to PF collection, to access changes induced by MI, these are the best control patients, which share main clinical characteristics and comorbidities also present in the MI cohorts. In the discussion we have introduced a section highlighting the limitations of the study (line 371-380) and reinforcing the lack of knowledge on the miR composition of the PF from healthy individuals (detailed in the next comment).

- The small sample size is a great limitation of this paper. Please discuss such a point in a dedicated limitation section.

We thank the reviewer for this suggestion, a limitation section was added at the end of the discussion (line 371-380): “A significant limitation of this study is the small sample size, collected for a single hospital. As such, future multicentered studies using larger cohorts are necessary to increase reliability and generalizability of our findings. Other limitation of this study is the present lack of knowledge on the miR composition of PF from healthy individuals to be used as a reference control. In fact, collection of PF from healthy patients is not possible because PF collection requires access to the pericardial space, only possible during open-chest surgery. As such, herein, stable coronary patients without previous history of MI were used as control which allows the identification of miR alterations upon acute MI but limits conclusions regarding the interest of the identified miRNA as biomarkers of coronary artery disease.

-Furthermore, a post-hoc sample size calculation should be provided.

Considering the miR-32-3p increase as primary outcome a post-hoc power of 97% versus NSTEMI and 74.2% versus control is obtained. As an alternative to post-hoc power, analysis of the width and magnitude of the 95% confidence interval (95% CI) may be a more appropriate method of determining statistical power. In such a case, CI for miR-32-3p differences are 0.74 ± 0.55 for the CTRL, 0.43 ± 0.24 for NSTEMI and 1.95 ± 0.73, further supporting our conclusions.

-Data related to CABG procedure should be discussed. ST2, for instance, is also an inflammatory marker as it is related to IL-33 (see also Ciccone MM et al. Molecules. 2013 Dec 11;18(12):15314-28): how can the time of extracorporeal circulation impact on the aims of the study? And the type of surgery? Please discuss all of these aspects.

We thank the reviewer for bringing these interesting topics that can be assessed in future work. Yet, in this particular work no effect is anticipated of either the type of surgery nor of the time of extracorporeal circulation once the PF is collected in the beginning of the surgery, immediately after opening the pericardial sac.

-Decimals deserve dots rather than virgules. Please revise the entire manuscript.

The manuscript has been revised according to the Reviewer suggestion. -

Reviewer 3 Report (New Reviewer)

Comments and Suggestions for Authors

This manuscript is an original study. Its primary focus is pericardial fluid miRNAs in patients with stable coronary artery disease and a history of acute myocardial infarction (STEMI and NSTEMI). The topic is modern and of interest, given the current trend to discover new cardiovascular biomarkers.

I have some minor observations:

 Lines 80-89. This text section discusses the results of the current study. This part needs to be moved/deleted. The Introduction should be expanded with more data on previous miRNA studies in PF and the known associations with disease/pathological processes (such as inflammation and fibrosis) should be commented.

Clarifications are needed for table 2.

- what does Antiaggregant mean? Oral antiplatelet therapy includes aspirin and P2Y12 receptor inhibitors. The two pharmacological classes should be mentioned in this way. Were there no patients to receive iSGLT2?

How do you comment on the small number of patients treated with ACE inhibitors or spironolactone after myocardial infarction, given the known benefits on fibrosis?

How were patients with myocardial infarction treated at the time of the index event? PCI? This aspect is important to mention considering the difference between patients with/without myocardial infarction in terms of LV ejection fraction.

Lines 238–239 should be consistent with lines 74–75 (here, we hypothesize that miRNA content of PF is altered upon impact of MI on cardiac fibroblast activation). Therefore, I suggest that STEMI be replaced by MI.

The Discussion section is very well written.

Thank you!

Comments on the Quality of English Language

English is generally good. 

Author Response

 Lines 80-89. This text section discusses the results of the current study. This part needs to be moved/deleted. The Introduction should be expanded with more data on previous miRNA studies in PF and the known associations with disease/pathological processes (such as inflammation and fibrosis) should be commented.

We acknowledge the Reviewer suggestions and the introduction was revised accordingly.

Clarifications are needed for table 2.

- what does Antiaggregant mean? Oral antiplatelet therapy includes aspirin and P2Y12 receptor inhibitors. The two pharmacological classes should be mentioned in this way. Were there no patients to receive iSGLT2?

We thank the reviewers for raising these questions as it was not clear that the medication information was related to the time of the surgery and the legend of Table 2 was revised to clarify this. In fact, although patients are under dual-antiplatelet therapy, owing to hemorragic risk, P2Y12 receptor inhibitors are suspended before surgery. Then, if the hemorragic risk is superior to ischemic risk, aspirin is also suspended in a sub-group of patients.

How do you comment on the small number of patients treated with ACE inhibitors or spironolactone after myocardial infarction, given the known benefits on fibrosis?

As mentioned in the previous comment the medication in Table 2 relates to the medication at the time of the surgery. In many cases, ACE inhibitors are stopped before CABG to avoid the potential deleterious consequences of perioperative hypotension.

How were patients with myocardial infarction treated at the time of the index event? PCI? This aspect is important to mention considering the difference between patients with/without myocardial infarction in terms of LV ejection fraction.

None of the patients were eligible to PCI and therefore this is not a variable to be considered in this study.

Lines 238–239 should be consistent with lines 74–75 (here, we hypothesize that miRNA content of PF is altered upon impact of MI on cardiac fibroblast activation). Therefore, I suggest that STEMI be replaced by MI.

In accordance with the reviewer's suggestions, this sentence was altered in the revised manuscript.

The Discussion section is very well written.

We truly thank the reviewer for the positive appreciation.

Reviewer 4 Report (New Reviewer)

Comments and Suggestions for Authors

Comments:

1. In the introduction, please add in background information of the difference between NSTEMI and STEMI, and the demonstration in terms of severity of symptom and survival rate. Also need to discuss the limitation of the control group used in this study.

2. In Figure 2G and H. NSTEMI seems to be less pro-fibrotic than the other two groups, what mir could be involved in or responsible for this result?

3. Which type of the cardiac cells are the source of the miRs?

Comments on the Quality of English Language

Please check the maintext for typo and grammar issue. 

Author Response

  1. In the introduction, please add in background information of the difference between NSTEMI and STEMI, and the demonstration in terms of severity of symptom and survival rate. Also need to discuss the limitation of the control group used in this study.

We thank the reviewer for the suggestion. A brief sentence was added to the introduction (line 48-55) and a brief discussion about the limitations of the study was included in the revised manuscript (line 371-380).

  1. In Figure 2G and H. NSTEMI seems to be less pro-fibrotic than the other two groups, what mir could be involved in or responsible for this result?

Although we do not have a clear explanation for these findings, we hypothesize that certain molecules present in NSTEMI may protect against myofibroblast differentiation. These molecules are not up-regulated in CTRL and STEMI, such as miR-16-5p, which is known to protect against fibroblast activation. However, we considered this information somehow speculative and therefore was not included in the manuscript but will be of interest to pursuit in future studies.

  1. Which type of the cardiac cells are the source of the miRs?

New experimental information showing that cardiac fibroblasts, cardiomyocytes and cardiac endothelial cells contribute to miRNA-22-3p production were introduced in the revised manuscript as Supplementary Figure 2C (line 241-249).

Round 2

Reviewer 1 Report (Previous Reviewer 3)

Comments and Suggestions for Authors

Dear Authors,

as already explained in my former review the complete study design is not clear and this could not be properly corrected although the main confusion is now claryfied. The fact that this clarification does not change anything on the general study suggests that this is not  a meaningful study, as it is irrelevant what happens in vitro. 

Author Response

The study design has been presented in the revised manuscript as Figure 1, in which a schematic representation of the study design is detailed. This image shows that pericardial fluid (PF) was collected from coronary patients undergoing CABG and subjected to in vitro function assays to evaluate cardiac fibroblasts activation, quantification of fibrosis-associated biomarkers and to small-RNA sequencing of which miR-22-3p was validated and tested in vitro regarding hCF activation.

Reviewer 2 Report (New Reviewer)

Comments and Suggestions for Authors

Authors well addressed my previous comments. The paper improved.

Author Response

Thank you very much for all the constructive criticism.

best regards,

This manuscript is a resubmission of an earlier submission. The following is a list of the peer review reports and author responses from that submission.

Round 1

Reviewer 1 Report

Comments and Suggestions for Authors

In this study pericardial fluid was collected patients undergoing coronary artery bypass grafting: control stable coronary artery patients, NSTEMI  and STEMI patients. The biological effect of the PF was assessed on human cardiac fibroblasts. The authors found that pericardial fluid after STEMI potentiates higher levels of fibroblast-to-myofibroblast activation in vitro and displays an accumulation of miRs associated with pro- and anti-fibrotic effects. 

I congratulate the authors for conducting this complex study. 

I must admit I find strange the lack of difference in patient and clinical characteristics between stable patients, patients with NSTEMI and STEMI. Usually these are groups with very different baseline risk.

I would appreciate if the authors could add a little bit more information on the clinical implication of their findings and how it may change future clinical practise and treatment approach.

Author Response

I congratulate the authors for conducting this complex study.

We truly acknowledge the Reviewer for the recognition of our work.

I must admit I find strange the lack of difference in patient and clinical characteristics between stable patients, patients with NSTEMI and STEMI. Usually these are groups with very different baseline risk.

We totally understand the point raised by the Reviewer as these patient groups normally present very different disease manifestations. However, the patients used in this study are from a surgical cohort, contributing to a certain degree of normalization amongst groups: NSTEMI cases selected for surgery are normally more severe situations in which patients are not illegible for percutaneous intervention and, on the other way, STEMI patients elected for CABG are not the most severe cases which require immediate intervention.

I would appreciate if the authors could add a little bit more information on the clinical implication of their findings and how it may change future clinical practise and treatment approach.

We thank the Reviewer for this constructive suggestion. To address this, two new sentences were included:

  • in the Introduction “In addition, whilst strategies for targeting therapeutics to the heart are scarce, intra-pericardial delivery route to reach cardiac cells has been emerging as a valuable alternative strategy. As such, studies unveiling the composition of the PF and how it impacts on cardiac cells in specific disease scenarios, are needed to establish the foreground knowledge for future clinical application.”.

  • In the Discussion “As such, this study contributes to increase our knowledge on the composition and modulatory role of PF after MI. This is particularly relevant to guide future interventions since the development of novel delivery technologies for non-invasive access to the pericardial sac in recent years has allowed the exploitation of the intra-pericardial delivery as an alternative route for local delivery of novel therapeutic agents and bioengineered solutions to the heart.”

Reviewer 2 Report

Comments and Suggestions for Authors

The reviewer thanks the authors for their interesting and unique submission. Unfortunately, the abstract and introduction are written sub-optimally and do not allow for me to complete my initial evaluation of the work. Please see my below queries and revise accordingly:

1. The abstract is at time non-sensical. Please carefully review and have a native English speaker complete the final grammar check please

2. What was the hypothesis for this body of work?

3. Reading the abstract and introduction produce more questions than answers. For example, it is very difficult to understand the groups/cohorts that were employed for this study. Subsequently, it is nearly impossible to discern the significant of the results. Please revise.

4. A central figure/illustration to summarize the details of the study design would be ideal.

5. The authors note that, "Even today, most of the knowledge regarding the pathophysiology of MI and the underlying mechanisms relies on the use of animal models [12,13] which, albeit useful in the preliminary stages of uncovering the mechanistic behind the pathology, seldom represents the human counterpart in all faithfulness." However the authors' present study is completed in an in-vitro fashion, which holds even less clinical relevance than in-vivo studies completed on animal models. The authors should carefully consider their justifications, otherwise the validity of their study comes into question with the very same concept.

6. Furthermore, please see the below reference regarding the use of human induced-pluripotent stem cell-derived cardiac myocytes to assess the molecular signals relevant in the pathophysiology of MI and possible paracrine/pericardial anti-fibrotic signaling.

Chinyere IR, Bradley P, Uhlorn J, Eason J, Mohran S, Repetti GG, Daugherty S, Koevary JW, Goldman S, Lancaster JJ. Epicardially Placed Bioengineered Cardiomyocyte Xenograft in Immune-Competent Rat Model of Heart Failure. Stem Cells Int. 2021 Jul 24;2021:9935679. doi: 10.1155/2021/9935679. PMID: 34341667; PMCID: PMC8325579.

7. Figure 5 is anatomically in correct, please revise or clarify the cardiac chambers and vessels.

8. Please specify the exact details regarding pericardial fluid sample acquisition. Please provide photographs to support your methodology.

9. Please speak to the implications of your work if the myocardial infarction is physically intra-myocardial or sub-endocardial, given the lack of contact with pericardial fluid.

10. It is not clear to this reviewer how the authors can claim that "We demonstrated that PF accumulates fibrotic molecules that correlate with functional parameters after MI," given that no negative/healthy controls were tested (if my understanding of the study design is correct). It may very well be the case that the same milieu of "fibrotic molecules" would be found in healthy control pericardial fluid.

Comments on the Quality of English Language

Please revise the entire manuscript for clarity and grammatical correctness, with a particular focus on the abstract and methods.

Author Response

1. The abstract is at time non-sensical. Please carefully review and have a native English speaker complete the final grammar check please

In accordance with the Reviewer suggestion, the revised manuscript has been altered to improve clarity and correct grammatical errors, particularly in the introduction, abstract and methods sections.

2. What was the hypothesis for this body of work?

The authors agree with the reviewer that the way our manuscript was presented in the abstract and in the introduction was unclear and we reformulated the last paragraph of the introduction clarify the hypothesis: “Herein, we hypothesize that the miRNA content of PF is altered upon MI impacting on the activation of cardiac fibroblasts. “

3. Reading the abstract and introduction produce more questions than answers. For example, it is very difficult to understand the groups/cohorts that were employed for this study. Subsequently, it is nearly impossible to discern the significant of the results. Please revise. A central figure/illustration to summarize the details of the study design would be ideal.

We sincerely appreciate the valuable feedback from the Reviewer. To improve clarification, the abstract and introduction were reformulated and a scheme illustrating the strategy employed in this study has been included in the manuscript (Figure 1).

4. The authors note that, "Even today, most of the knowledge regarding the pathophysiology of MI and the underlying mechanisms relies on the use of animal models [12,13] which, albeit useful in the preliminary stages of uncovering the mechanistic behind the pathology, seldom represents the human counterpart in all faithfulness." However, the authors' present study is completed in an in-vitro fashion, which holds even less clinical relevance than in-vivo studies completed on animal models. The authors should carefully consider their justifications, otherwise the validity of their study comes into question with the very same concept.

We thank the reviewer for raising this point, and we totally agree. As a result, the mentioned paragraph was removed from the manuscript.

5. Furthermore, please see the below reference regarding the use of human induced-pluripotent stem cell-derived cardiac myocytes to assess the molecular signals relevant in the pathophysiology of MI and possible paracrine/pericardial anti-fibrotic signaling.

Chinyere IR, Bradley P, Uhlorn J, Eason J, Mohran S, Repetti GG, Daugherty S, Koevary JW, Goldman S, Lancaster JJ. Epicardially Placed Bioengineered Cardiomyocyte Xenograft in Immune-Competent Rat Model of Heart Failure. Stem Cells Int. 2021 Jul 24;2021:9935679. doi: 10.1155/2021/9935679. PMID: 34341667; PMCID: PMC8325579.

We thank the Reviewer for the suggestion. A brief discussion about this subject was included in the revised manuscript: “Furthermore, the impact of PF on other cardiac cells such as cardiomyocytes and even in intercellular communication involving, for instance, cardiomyocytes and cardiac fibroblasts is further needed and can be achieved resorting to human induced pluripotent stem cells. Investigating this combination may unveil crucial molecular signals pertinent to the pathophysiology of MI.”

6. Figure 5 is anatomically in correct, please revise or clarify the cardiac chambers and vessels.

We thank the reviewer for the comment and, in the revised manuscript, Figure 5 (now Figure 6) was edited accordingly.

7. Please specify the exact details regarding pericardial fluid sample acquisition. Please provide photographs to support your methodology.

In accordance with the Reviewer's suggestion, and to clarify sample collection, we have added to the revised manuscript a video (Video 1) demonstrating the procedure for PF collection during CABG.

8. Please speak to the implications of your work if the myocardial infarction is physically intra-myocardial or sub-endocardial, given the lack of contact with pericardial fluid.

We acknowledge the reviewer for this suggestion, which was addressed in the Discussion by adding the following: “Other aspect that may influence the accumulation of cardiac miR in the PF is the extent and localization of MI. For example, one would expect that a in sub-endocardial MI, common in NSTEMI patients, would have a smaller diffusion to the PF when compared to large transmural MI, more common in STEMI. Future studies associating biomarker quantification in the PF with infarct size and transmurality obtained by magnetic resonance imaging may provide further clarification.”

9. It is not clear to this reviewer how the authors can claim that "We demonstrated that PF accumulates fibrotic molecules that correlate with functional parameters after MI," given that no negative/healthy controls were tested (if my understanding of the study design is correct). It may very well be the case that the same milieu of "fibrotic molecules" would be found in healthy control pericardial fluid.

The authors would like to express their appreciation for the review comment. We have now clarified the studied cohorts (revised manuscript, Figure 1). As such, a control group consisting of individuals with stable coronary artery disease and no history MI was used to understand possible differences induced by MI. Yet, we fully agree with the reviewer that the text was not clear nor was accurately describing the observed results, and therefore this sentence was reformulated. Our analysis revealed an inverse correlation between the fibrotic molecule ST-2 and left ventricle ejection fraction (LVEF) specifically in STEMI patients. Notably, no such correlation was observed in the other groups. In the revised version of the manuscript we provide the results showing the lack of correlation between LVEF and ST-2 levels in PF of NSTEMI and CTRL patients (Supplementary Figure 1A,B).

Please revise the entire manuscript for clarity and grammatical correctness, with a particular focus on the abstract and methods.

The manuscript has been revised according to the Reviewer suggestion.

Reviewer 3 Report

Comments and Suggestions for Authors

In the current study the authors investigated the biological potential of pericardial fluid components on cardiac remodeling.

Major comments:

It is rather difficult to follow the authors during this manuscript. In the abstract they claimed a couple of main findings that are not really represent the data of the study.

First, “PF accumulates fibrotic and inflammatory molecules that correlate with functional parameters after MI.” I guess you mean PICP and ST2 (Fig. 1). If yes, is ST2 indeed what is also known as IL1RL1? Within your study you call the factor serum stimulation-2? This is however, neither a protein nor a gen name for this molecule. Sometimes you write ST2 but then ST-2. Is this the same? In anyway, you correlated only ST2 with function and therefore the statement is not correct as it stands in the abstract.

Second, “The PF from STEMI patients potentiates higher levels of fibroblast-to-myofibroblast activation in vitro…”.  I guess you claim the ACTA2 experiment of Fig. 1E. If yes, you might have noticed that the statistical power of this effect is rather low and not seen in Fig. 1G. The rational for this statement is not clear!

Third, “…and displays an accumulation of 26 miRs associated with pro- and anti-fibrotic effects.” It is not clear how you come to this conclusion. The term ‘cardiac remodeling’ (Fig. 2E) is brought.  Mainly, there seems to be a connection to inflammation (NF-kB, T-Cell Activation etc.).

Fourth, “From those, miR-22-3p was found to inhibit TGF-β-induced human cardiac fibroblast activation in vitro…”. This is indeed the case but the summary figure brings confusion again. The only known function of miRs is to reduce mRNA stability. In Fig. 5 you give the impression that the extracellular protein is blocked. This can hardly by the case.

Fifth, “…supporting a potential effect on counterbal- 28 ancing fibrogenic molecules present in PF following extensive MI.” This statement is simply speculative. Your data show the opposite. Tthe STEMI patients have the largest pro-fibrotic effect.

Minor points

Fig. 3 mislabled

CAL1A1 is the gen-name. Please use Collagen-1 for your histological slights.

Table 2: Please use ‘NSTEMI’ instead of MI as STEMI and NSTEMI are MIs.

Author Response

Major comments:

It is rather difficult to follow the authors during this manuscript. In the abstract they claimed a couple of main findings that are not really represent the data of the study.

We sincerely appreciate the valuable feedback provided by the Reviewer. The manuscript has been revised for coherence, clarity and grammar, including the abstract.

First, “PF accumulates fibrotic and inflammatory molecules that correlate with functional parameters after MI.” I guess you mean PICP and ST2 (Fig. 1). If yes, is ST2 indeed what is also known as IL1RL1? Within your study you call the factor serum stimulation-2? This is however, neither a protein nor a gen name for this molecule. Sometimes you write ST2 but then ST-2. Is this the same? In anyway, you correlated only ST2 with function and therefore the statement is not correct as it stands in the abstract.

The authors acknowledge the Reviewer for bringing this to our attention. The revised manuscript has been revised accordingly, namely we have now included the full name of the soluble protein, soluble suppression of tumorigenicity 2 (ST-2), normalized the abbreviation of the molecule and reformulated the abstract to provide clarity regarding the proteins that were correlated in our study.

Second, “The PF from STEMI patients potentiates higher levels of fibroblast-to-myofibroblast activation in vitro…”.  I guess you claim the ACTA2 experiment of Fig. 1E. If yes, you might have noticed that the statistical power of this effect is rather low and not seen in Fig. 1G. The rational for this statement is not clear!

The authors thank the reviewer for this comment and revised the manuscript accordingly. The authors agree with the reviewer that the way how these results were presented could be misleading and we reformulated the introduction and the abstract to better reflect the possible impact of the PF in cardiac fibroblasts for the following “PF of the three groups induces similar levels of fibroblast-to-myofibroblast activation in vitro, despite that higher production of Collagen type I was found in response to STEMI PF.”

Third, “…and displays an accumulation of 26 miRs associated with pro- and anti-fibrotic effects.” It is not clear how you come to this conclusion. The term ‘cardiac remodeling’ (Fig. 2E) is brought.  Mainly, there seems to be a connection to inflammation (NF-kB, T-Cell Activation etc.).

We thank the reviewer for this comment. The gene enrichment analysis for miR function highlighted Cardiac remodeling as the GO term with lowest p-value (now Fig.3E). Further analysis of these miRs showed that 8/9 miRNA been previously implicated on cardiac fibroblast activation, either by promoting or inhibiting myofibroblast formation, for which we used the designation of “pro- and anti-fibrotic”. Please note, that the original work supporting this effect for each miR have been included in the sentence.

Fourth, “From those, miR-22-3p was found to inhibit TGF-β-induced human cardiac fibroblast activation in vitro…”. This is indeed the case but the summary figure brings confusion again. The only known function of miRs is to reduce mRNA stability. In Fig. 5 you give the impression that the extracellular protein is blocked. This can hardly by the case.

We acknowledge and appreciate the constructive feedback from the reviewer. In line with the suggested revisions Fig 5 (now Fig 6) has been revised accordingly.

Fifth, “…supporting a potential effect on counterbal- 28 ancing fibrogenic molecules present in PF following extensive MI.” This statement is simply speculative. Your data show the opposite. Tthe STEMI patients have the largest pro-fibrotic effect.

We thank the reviewer for highlighting this and we decided to remove this speculative sentence from the manuscript.

Minor points

Fig. 3 mislabled

The figure 3, now figure 4, has been revised accordingly.

CAL1A1 is the gen-name. Please use Collagen-1 for your histological slights.

Corrected accordingly, we changed the gene name to the protein name when adequate and used “Type I collagen” as protein name in the text and the abbreviation COL I in the images.

Table 2: Please use ‘NSTEMI’ instead of MI as STEMI and NSTEMI are MIs.

We have followed the reviewer suggestions and revised tables and images with the correct designation.

Round 2

Reviewer 1 Report

Comments and Suggestions for Authors

Authors have adressed all of the raised comments and the manuscript has imporved substiantially.

Author Response

We would like to thank you for the comments, and we are happy that we addressed all of them satisfactorily.

Reviewer 2 Report

Comments and Suggestions for Authors

The reviewer thanks the authors for their revisions and improvements to the articles, as well as their thorough point-by-point response to this reviewer's critiques.

0. Please provide an accepted-track changes version, to improve the readability of the extensively revised article.

1. Please consider improving the title of the article to better capture the novelty of the article.

2. Please remove the remaining open parenthesis (line 29), and perform careful proof-reading of the revised manuscript with an accepted-track changes version.

3. Please provide this reviewer ample evidence, in the form of references to high-quality original articles, that supports the claim in the abstract (line 29) regarding pericardial microRNAs being "critical" modulators of the cardiac pathologic response post-myocardial infarction.

The reviewer would like to remind the authors that the scientific meaning of the word "critical" confers 100%, completely, and unequivocally necessary for the related biologic process to occur. It is not clear to this reviewer that "...microRNAs...are critical modulators of the cardiac pathologic response" as it relates to acute myocardial infarction, however perhaps this claim is true for pericarditis. The reviewer would caution the authors to rephrase or redact, and appropriately soften any remaining claims in the article that are potentially contentious.

4. While the reviewer appreciates the addition of Video 1, the video cannot be opened by this reviewer from the PDF. Perhaps the authors can 1) send the video file with the supplementary word document, 2) provide a hyperlink to the video file privately hosted on the internet, and/or 3) provide a text-based detailed description of the process for pericardial fluid acquisition to supplement the video.

5. The reviewer thanks the authors for their additions regarding the use of human induced-pluripotent stem cell-derived cardiac myocytes to assess the molecular signals relevant in the pathophysiology of MI and possible paracrine/pericardial anti-fibrotic signaling, and thank the authors for including the provided reference as evidence.

The reviewer would like to call special attention to the fact that the new reference 54 is not highlighted to communicate any track-changes. All changes to the manuscript from the initial submission should be denoted. This improves the clarity and quality of the peer review process. Please ensure that any and all changes to the references list, in addition to the body of the manuscript, are clearly denoted.

Author Response

The reviewer thanks the authors for their revisions and improvements to the articles, as well as their thorough point-by-point response to this reviewer's critiques.

  1. Please provide an accepted-track changes version, to improve the readability of the extensively revised article.

We have followed the IJMS instructions which requests a revised version of the manuscript with track-changes. We fully agree that, in a very altered manuscript, it may be difficult to follow the alterations. As such, we have included a revised version with accepted changes and yellow and green highlights on the text that was revised in revision round 1 and 2, respectively, to improve readability of the manuscript.

  1. Please consider improving the title of the article to better capture the novelty of the article.

The authors acknowledge the suggestion and the title was changed accordingly to: Pericardial fluid accumulates microRNAs that regulate heart fibrosis after myocardial infarction

  1. Please remove the remaining open parenthesis (line 29), and perform careful proof-reading of the revised manuscript with an accepted-track changes version.

We appreciate the observation; Proof-reading was done in the whole document to correct small mistakes in the text.

  1. Please provide this reviewer ample evidence, in the form of references to high-quality original articles, that supports the claim in the abstract (line 29) regarding pericardial microRNAs being "critical" modulators of the cardiac pathologic response post-myocardial infarction.

The reviewer would like to remind the authors that the scientific meaning of the word "critical" confers 100%, completely, and unequivocally necessary for the related biologic process to occur. It is not clear to this reviewer that "...microRNAs...are critical modulators of the cardiac pathologic response" as it relates to acute myocardial infarction, however perhaps this claim is true for pericarditis. The reviewer would caution the authors to rephrase or redact, and appropriately soften any remaining claims in the article that are potentially contentious.

We recognize that using the word “critical” is not the most adequate and changed it to the word “important” to reflect that, although microRNAs are strongly involved in the onset and progression of cardiac fibrosis, they might not be strictly necessary for this process to occur.

  1. While the reviewer appreciates the addition of Video 1, the video cannot be opened by this reviewer from the PDF. Perhaps the authors can 1) send the video file with the supplementary word document, 2) provide a hyperlink to the video file privately hosted on the internet, and/or 3) provide a text-based detailed description of the process for pericardial fluid acquisition to supplement the video.

We appreciate the comment, the video will be provided as a stand-alone supplementary file to allow for easier access to the material (Supplementary Video 1) and in case problems are found we will request assistance from the editorial team.

  1. The reviewer thanks the authors for their additions regarding the use of human induced-pluripotent stem cell-derived cardiac myocytes to assess the molecular signals relevant in the pathophysiology of MI and possible paracrine/pericardial anti-fibrotic signaling, and thank the authors for including the provided reference as evidence.

We appreciate the reviewer reference suggestion, and we are positive that it enriched the overall discussion of the manuscript.

The reviewer would like to call special attention to the fact that the new reference 54 is not highlighted to communicate any track-changes. All changes to the manuscript from the initial submission should be denoted. This improves the clarity and quality of the peer review process. Please ensure that any and all changes to the references list, in addition to the body of the manuscript, are clearly denoted.

Reviewed as suggested.

Reviewer 3 Report

Comments and Suggestions for Authors

Dear Authors,

I thank you for the changes that you performed. Nevertheless, I am not convinced that this study improves our knockledgt substanially. In fact, most parts are very speulative. If the aim of the study is a focus on miRs and you identify finally only one specific miR then you must identify the molecular targets of this miR. The summary shows an arrow on CMC (means activation?) and an inhbition of TGFß on fibroblasts. The text suggests another way around. I still do not get an idea what you want to suggest and what are the experiments that support this.

Author Response

I thank you for the changes that you performed. Nevertheless, I am not convinced that this study improves our knockledgt substanially. In fact, most parts are very speulative.

The aim of this study was to evaluate the impact of PF on cardiac fibroblast activation and identify altered miR content in the PF of STEMI patients with potential to integrate anti-fibrotic therapies for MI. To address this, we combined RNA-sequencing with in vitro models of human cardiac fibroblast-to-myofibroblast differentiation.

The first original contribution of our work is to demonstrate that PF accumulates higher concentrations of fibrotic molecules PICP and ST-2 being the latter correlated with systolic dysfunction in STEMI patients. These findings support the relevance of using the PF as a valuable sources of biomarkers disease severity and predictors of cardiovascular complications. The second original contribution of our work relies of the demonstration that PF of STEMI patients is enriched in miRNAs with dual effect on fibrinogenesis, which points to an active role of this biofluid in the regulation of cardiac fibrosis. Finally, the third original contribution of this work is to show that miR-22-3p which is enriched in STEMI patients and that it inhibits human fibroblast-to-myofibroblast activation in vitro. As such, this study contributes to increase our knowledge on the composition and modulatory role of PF after MI. This is particularly relevant to guide future interventions since the development of novel delivery technologies for non-invasive access to the pericardial sac in recent years has allowed the exploitation of the intra-pericardial delivery as an alternative route for local delivery of novel therapeutic agents and bioengineered solutions to the heart.

Aspects that could be considered more speculative were revised and removed from the text.

If the aim of the study is a focus on miRs and you identify finally only one specific miR then you must identify the molecular targets of this miR.

We demonstrated that miR-22-3p overexpression in human cardiac fibroblasts inhibited fibroblast-to-myofibroblast differentiation. Indeed, in murine cardiac fibroblasts, miR-22-3p was found to protect from TGF-β-mediated activation [56], a mechanism that was attributed to direct inhibition of TGF-βR1 [64]. We have knowledged these works on the discussion as have already identified the the molecular targets of miR-22-3p on the TGF-β pathway.

The summary shows an arrow on CMC (means activation?) and an inhbition of TGFß on fibroblasts. The text suggests another way around. I still do not get an idea what you want to suggest and what are the experiments that support this.

The scheme shows that miRNA-32-3p, accumulated in the PF after MI, may impact on myocardial resident fibroblasts by reducing fibroblast-to-myofibroblast differentiation as we demonstrated in vitro.

Round 3

Reviewer 2 Report

Comments and Suggestions for Authors

The reviewer thanks the authors for their thorough revisions.